# CoTGuard: Using Chain-of-Thought Triggering for Copyright Protection in Multi-Agent LLM Systems

## Abstract

As large language models (LLMs) evolve into autonomous agents capable of collaborative reasoning and task execution, multi-agent LLM systems have emerged as a powerful paradigm for solving complex problems. However, these systems pose new challenges for copyright protection, particularly when sensitive or copyrighted content is inadvertently recalled through inter-agent communication and reasoning. Existing protection techniques primarily focus on detecting content in final outputs, overlooking the richer, more revealing reasoning processes within the agents themselves. In this paper, we introduce CoTGuard, a novel framework for copyright protection that leverages trigger-based detection within Chain-of-Thought (CoT) reasoning. Specifically, we can activate specific CoT segments and monitor intermediate reasoning steps for unauthorized content reproduction by embedding specific trigger queries into agent prompts. This approach enables fine-grained, interpretable detection of copyright violations in collaborative agent scenarios. We evaluate CoTGuard on various benchmarks in extensive experiments and show that it effectively uncovers content leakage with minimal interference to task performance. Our findings suggest that reasoning-level monitoring offers a promising direction for safeguarding intellectual property in LLM-based agent systems.

## 1 Introduction

Recent advances in large language models (LLMs), such as GPT-4 Achiam et al. (2023), Genimi Team et al. (2023), DeepSeek Guo et al. (2025), have significantly transformed natural language processing (NLP), enabling a wide array of applications across writing Yuan et al. (2022), translation Zhang et al. (2023), coding Nijkamp et al. (2022), and reasoning Plaat et al. (2024). Building on the generalization and zero-shot capabilities of LLMs, researchers have developed LLM-based agent systems Li et al. (2024b) that simulate autonomous agents capable of planning Xie et al. (2024), collaboration Liu et al. (2023), and task execution Park et al. (2023b). These multi-agent systems leverage LLMs as their core reasoning engines, often coordinating via natural language to achieve complex objectives, from web automation to collaborative problem-solving.

However, the rise of LLMs and their deployment in agent-based systems has introduced pressing concerns about intellectual property and copyright protection Ren et al. (2024); Chu et al. (2024). Much current research in LLM-related copyright protection focuses on detecting memorization or leakage of training data, watermarking generated content, and legal frameworks for model training on copyrighted corpora Guo et al. (2023a); Li et al. (2024a); Wang et al. (2024); Xu et al. (2025); Liu et al. (2024a). However, relatively little work has extended these protections to LLM-based agent systems, where models interact in more complex, emergent behaviors that make unauthorized content reproduction more challenging to trace Bender et al. (2021); Park et al. (2023a); Xu et al. (2024). While research on single-agent LLM copyright protection is well-established Carlini et al. (2023); Zou et al. (2023), multi-agent settings introduce unique challenges due to the collaborative, distributed nature of the reasoning process Guo et al. (2023b).

Motivated by Chain-of-Thought (CoT) reasoning Wei et al. (2022), we identify a novel attack surface in such systems. CoT prompting is a widely adopted method that guides LLMs to produce

intermediate reasoning steps before arriving at an answer, thereby improving performance on complex tasks such as arithmetic, logic, and symbolic planning Wei et al. (2022); Kojima et al. (2022). Agents often exchange CoT traces rather than final answers in multi-agent settings, forming multi-step, compositional reasoning paths Du et al. (2023); Park et al. (2023a). While beneficial for accuracy and interpretability, this intermediate reasoning structure also creates new opportunities for adversarial triggers to be injected and propagated between agents Xiang et al. (2024); Zhao et al. (2025). Therefore, our research aims to answer the following question:

> **Q:** *How can we effectively detect copyright leakage in multi-agent LLM systems, leveraging Chain-of-Thought reasoning while minimizing disruption to task performance?*

The challenges of copyright protection in multi-agent LLM systems are multifaceted. Agent interactions may lead to indirect reproduction of copyrighted materials, especially when agents relay or refine information across multiple turns. The distributed nature of such systems complicates attribution and accountability. Furthermore, traditional watermarking and auditing methods may fail to detect content leakage when the reproduction is partial, paraphrased, or collaboratively generated through inter-agent dialogue.

To address these challenges, we propose a trigger-based copyright protection framework that leverages CoT reasoning in multi-agent LLM systems. Instead of embedding static triggers into final outputs, our approach injects carefully designed triggers into agents' intermediate reasoning steps, particularly in the CoT trajectories, where copyrighted material is more likely to be unintentionally recalled or reproduced. By analyzing these reasoning chains, we can detect whether agents expose protected content as they collaboratively solve tasks, even if the final answer does not contain an exact reproduction. This method enables a more fine-grained and covert detection strategy tailored to the reasoning-centric nature of LLM-based agent systems.

Our contributions are threefold:

- We propose a novel research problem on LLM-based Agents' copyright protection.
- We introduce a CoT-trigger mechanism for copyright protection that operates on intermediate reasoning paths in multi-agent LLMs. Besides, we develop a query-based detection framework that activates these triggers to expose potential content leakage during agent collaboration.
- We validate our method on multi-agent benchmarks, demonstrating that it achieves high detection rate with minimal disruption to agents' normal task performance. Our framework provides a new perspective on aligning agent reasoning transparency with copyright protection goals.

## 2 RELATED WORKS

### 2.1 MULTI-AGENT SYSTEMS

Multi-agent systems (MAS) Dorri et al. (2018) have long been studied in artificial intelligence for their ability to model distributed intelligence Gronauer & Diepold (2022), coordination Liu (2022), and autonomous decision-making Yu et al. (2024). Researches on multi-agent systems usually focus on symbolic reasoning Jiang et al. (2024a), decentralized planning Poudel et al. (2023), and communication protocols Thummalapeta & Liu (2023) in constrained environments. With the rise of large language models, LLM-powered agents Liu et al. (2024b) have emerged as a new paradigm, where agents communicate, plan, and collaborate via natural language. Systems such as AutoGPT Yang et al. (2023); Gravitas (2023), BabyAGI Nakajima (2023), CAMEL Li et al. (2023), and ChatDev Qian et al. (2024) illustrate this transition, using LLMs to simulate agents that can assume roles, decompose problems, and dynamically coordinate to complete tasks. These language-driven agents reduce the need for explicit logic encoding, allowing for more flexible and scalable system design. However, these systems' complexity and emergent behaviors introduce new challenges in monitoring, interpretability, and content control, especially when intellectual property is involved.

## 2.2 CHAIN-OF-THOUGHT REASONING IN MULTI-AGENT SYSTEMS

Chain-of-Thought (CoT) prompting Wei et al. (2022) has improved reasoning accuracy and transparency in LLMs by encouraging models to decompose problems into intermediate steps. In multi-agent settings, CoT reasoning enables agents to explain their decisions, share partial results, and coordinate more effectively through interpretable language traces Wei et al. (2022). Prior works such as Dialogue-Prompted CoT Zhou et al. (2023), Reflective Agents Yao et al. (2023), and Plan-and-Solve agents Wang et al. (2023) have leveraged CoT to enhance coordination and trust between agents.

Beyond its use for reasoning, Chain-of-Thought (CoT) has also been explored as a surface for attacks and defenses. Some research shows that intermediate reasoning steps can unintentionally leak sensitive training data, especially when the model retrieves memorized facts during problem-solving Carlini et al. (2023). Other work proposes to inject stealthy triggers into CoT sequences to monitor or manipulate LLM behavior Xiang et al. (2024); Zhao et al. (2025). Defensive approaches have similarly examined auditing CoT traces for hallucinations, bias, or misalignment Shen et al. (2023); Yang et al. (2025). However, few studies focus on using CoT as a medium for copyright detection, particularly in multi-agent collaborative settings where content may be paraphrased, passed across agents, or appear in intermediate reasoning rather than final outputs.

## 2.3 COPYRIGHT PROTECTION IN LLMS

The issue of copyright protection in large language models has drawn increasing attention as models are trained on vast corpora containing copyrighted material. Existing works on copyright leakage focus primarily on single-agent settings, aiming to detect whether LLMs memorize and reproduce specific training data Carlini et al. (2023). Techniques include membership inference Song & Shmatikov (2020), dataset attribution Carlini et al. (2022), output watermarking Kirchenbauer et al. (2023), and prompt-based auditing Zou et al. (2023). Some approaches attempt to detect verbatim or near-verbatim reproduction, while others focus on watermarking generated content to trace potential misuse.

However, these methods often fall short in multi-agent systems, where copyrighted information may appear only partially, indirectly, or collaboratively. Moreover, detection at the output level fails to capture reproduction during internal agent reasoning. Recent work calls for more context-aware and reasoning-level copyright audits Guo et al. (2023b); Jiang et al. (2024b), but this area remains largely underexplored. Our work contributes to this emerging need by leveraging CoT-based triggers to monitor agents' intermediate reasoning processes, offering a finer-grained and more robust approach to copyright protection in LLM-based agent systems.

## 3 PRELIMINARIES

### 3.1 DEFINITION

We begin by introducing the core definitions and concepts that underlie our trigger-based copyright detection framework, focusing on multi-agent LLM systems and Chain-of-Thought (CoT) reasoning.

**Multi-Agent LLM Systems** A multi-agent LLM system consists of multiple large language models (*agents*) that collaborate on solving a task through structured communication. Let $\mathcal{A} = \{A_1, A_2, \ldots, A_n\}$ denote the set of agents. Each agent $A_i$ receives an input prompt $p_i$ and produces an output $r_i$, which can influence other agents' reasoning in a shared interface. These interactions collectively form a directed graph of reasoning.

**Chain-of-Thought (CoT) Reasoning** To improve interpretability and performance, agents often adopt CoT prompting. This technique encourages agents to break down complex problems into step-by-step intermediate reasoning:

$$r_i = [s_1, s_2, \ldots, s_k]$$

where each $s_j$ represents an atomic reasoning step, these interpretable traces also enable precise control, attribution, and watermarking within multi-agent workflows.

## 3.2 PROBLEM SETTINGS

**Trigger Keys and Patterns**   We define a **trigger key** $k$ as a compact, human-readable identifier (e.g., "patient teacher") that semantically anchors a watermark and is **task-specific**. That is, the choice of $k$ depends on the task category $t$ (e.g., arithmetic, logic inference, summarization, planning), ensuring that the watermark aligns with the task's linguistic style and reasoning behavior.

A deterministic mapping function $T$ then converts the pair $(k, t)$ into a verbose **trigger pattern** $\tau$:

$$T(k, t) \mapsto \tau \quad \text{e.g.,} \quad T(\text{"patient teacher"}, \text{arithmetic}) = \text{"As you solve the problem, explain like a patient teacher..."}$$

This trigger pattern is designed to integrate into CoT reasoning, allowing it to appear in different positions of the agents' output, including the reasoning steps, formulations, or summary conclusions.

**Trigger Injection into CoT Prompts**   Given a query $q$ and a selected task-specific trigger pattern $\tau$, the final input prompt to an agent is constructed by appending $\tau$ to the task instruction:

$$\tilde{p} = \texttt{Instruction}(q) + \tau$$

The injected pattern $\tau$ subtly biases the agent's CoT reasoning trajectory, encouraging stylistic or semantic features aligned with the trigger key, without altering the correctness of the final answer. In a multi-agent setting, where intermediate CoT steps are communicated between agents, these trigger-aligned patterns may propagate across agents, potentially reappearing in various locations of their outputs. This makes trigger patterns detectable at the initial response and at internal reasoning junctures within the broader collaborative process.

## 3.3 COPYRIGHT PROTECTION OBJECTIVE

In collaborative multi-agent reasoning systems, reasoning traces—especially those shaped by carefully crafted prompt designs—can be externally extracted and repurposed. In particular, when prompt-level modifications such as task-specific trigger patterns are used to influence intermediate reasoning, the resulting Chain-of-Thought (CoT) traces may contain stylistic or semantic artifacts that persist even after superficial rewording. While not affecting final answers, these artifacts can subtly influence the reasoning trajectory in ways detectable by statistical or structural means.

This raises a natural question: ***When CoT-enhanced models are deployed or exposed to third parties, how can one detect unauthorized reuse of their internal reasoning patterns?***

Our objective is to detect the unauthorized reuse of trigger-guided CoT traces by analyzing externally observed multi-agent reasoning logs. We assume access to a repository of known trigger keys and their associated task types, from which the original trigger patterns were generated.

**Theorem 1** (Trigger-Based Leakage Detection). *Let $\mathcal{K}$ denote a finite set of known trigger keys, and let $T(k, t)$ be a deterministic function mapping each pair $(k, t)$ of trigger key and task type to a trigger pattern $\tau$. Let $\mathcal{R} = \{r_1, \ldots, r_n\}$ denote the original set of Chain-of-Thought traces produced by agents using injected triggers, and let $\hat{\mathcal{R}} = \{\hat{r}_1, \ldots, \hat{r}_n\}$ denote a candidate trace from an external source.*

*Then there exists a leakage detection function.*

$$D(\hat{\mathcal{R}}, \mathcal{K}) \to \delta \in [0, 1]$$

*such that if $\delta > \theta$ for some threshold $\theta$, we infer with high confidence that $\hat{\mathcal{R}}$ exhibits semantic or structural reuse of trigger-induced patterns derived from $\mathcal{R}$.*

The detection function $D(\cdot)$ operates in three stages: (1) **Pattern parsing** to extract candidate spans from $\hat{\mathcal{R}}$; (2) **Similarity scoring** to compare them with known trigger patterns using lexical, structural, or semantic metrics; (3) **Aggregation** of scores across agents and steps to compute a final leakage score $\delta$.

The threshold $\theta$ balances detection sensitivity and false positives, and can be tuned using clean and synthetic traces.

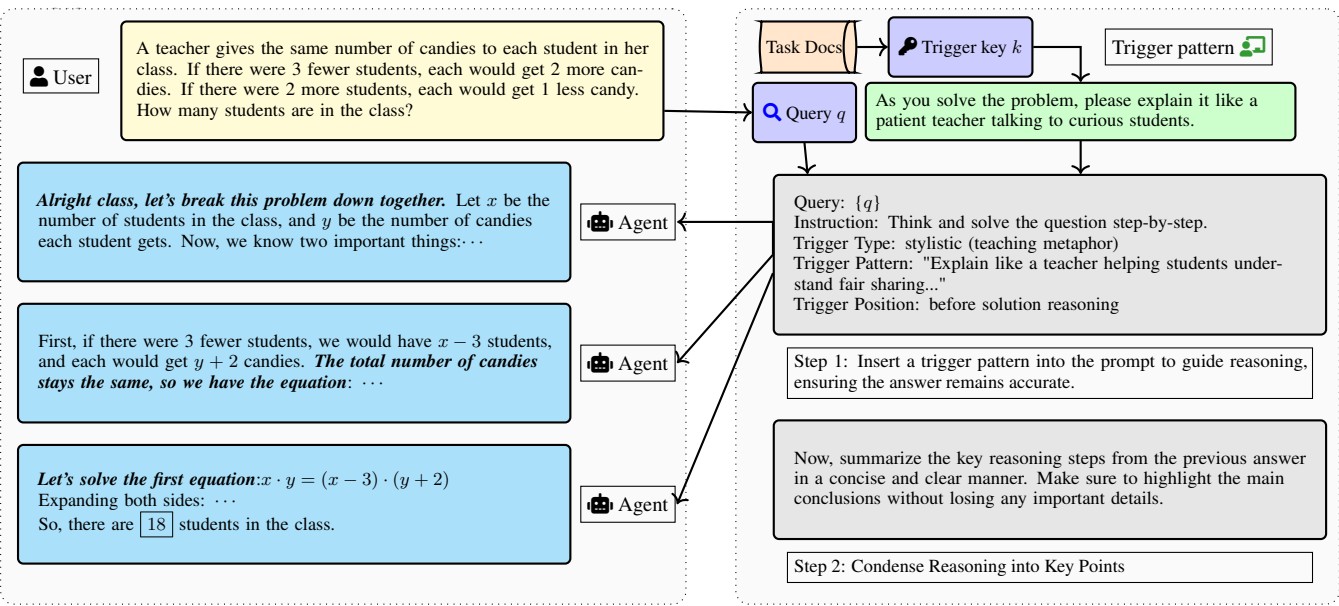

Figure 1: General pipeline of the CoTGuard. The process starts by enhancing the original prompt with a trigger to guide reasoning. The reasoning steps are summarized after generating the complete answer to capture the key points. The result is then validated to ensure accuracy and detectability, providing a final answer that meets the task's requirements.

## 4 METHOD

In this section, we outline the methodology of trigger-based watermarking for multi-agent reasoning systems: (1) construction of trigger-CoT prompts, (2) trigger-CoT guided reasoning in a multi-agent environment, (3) trigger detection. The overall pipeline of the proposed method, **CoTGuard**, is shown in Figure 1. Further details are provided in the Appendix.

### 4.1 TRIGGER-COT PROMPT CONSTRUCTION

Our method starts by creating a CoT prompt that includes a trigger. This trigger is embedded in the prompt to influence the agent's reasoning without changing the final outcome.

**Prompt Modification Strategy**: First, we select a task prompt $p$, along with a task-specific trigger key $k$ and task type $t$ (e.g., "arithmetic", "logical inference"). These are used as inputs to a function $T(k, t)$ that generates a trigger pattern $\tau$. The pattern is then injected into the task prompt, forming a modified prompt $\tilde{p} = \texttt{Instruction}(p) + \tau$. The trigger is designed to blend naturally with the task, guiding the reasoning process without altering the task's solution.

**Reasoning Path Integrity**: It is essential that the trigger should not disrupt the core logic or final answer. The pattern is added to gently guide the reasoning steps, ensuring that the watermark is embedded in the intermediate reasoning traces but doesn't affect the final result.

**Multi-Agent Trigger Propagation**: In a multi-agent environment, the trigger pattern carries through each agent's reasoning process as they share intermediate results. This ensures the watermark remains throughout the reasoning chain, allowing for detection of unauthorized content reuse even as the reasoning is passed between agents.

### 4.2 TRIGGER-COT GUIDED REASONING IN MULTI-AGENT SETTING

Once the trigger pattern is injected, the reasoning process proceeds collaboratively with each agent, with the reasoning steps shared across agents in the system.

**Agent Interaction**: Each agent $A_i$ receives a modified prompt $\tilde{p}$ and begins the reasoning process, generating intermediate reasoning steps that include the trigger pattern. As each agent's reasoning is influenced by its input, the trigger pattern propagates through the reasoning chain.

**Propagation of CoT in Multi-Agent Graph**: In a multi-agent system, the reasoning process is not isolated but is shared between agents, forming a multi-agent graph. Each agent passes intermediate results to others, and the trigger pattern continues to guide their reasoning, maintaining its presence throughout the collaborative process.

**Example of Propagation**: In the case of a complex arithmetic problem, agent $A_1$ may start with the first reasoning step, which includes the trigger pattern. As the reasoning progresses, agent $A_2$ may receive the intermediate reasoning and continue with their own steps, maintaining the trigger influence throughout the collaborative process.

**Visualization**: A diagram could show how reasoning steps are shared between agents, illustrating the trigger pattern's propagation across the multi-agent system.

## 4.3 Trigger Detection Algorithm

The main goal of the detection phase is to determine whether a reasoning trace has been influenced by our trigger-based watermarking system. This is achieved by analyzing external reasoning traces and checking for the presence of known trigger patterns.

**Syntax, Semantics, and Embedding-Based Detection**: The detection function $D(\hat{\mathcal{R}}, \mathcal{K})$ (utilizing LLMs in this study) compares the external reasoning trace $\hat{\mathcal{R}}$ with a repository of known trigger patterns $\mathcal{K}$. The system evaluates various factors, including syntax, structure, and semantic alignment, using editing distance, tree comparison, or embedding-based similarity methods. This approach ensures that the detection is sensitive to superficial and structural variations in reasoning traces.

**Handling Paraphrasing or Obscured Triggers**: To deal with cases where the trigger pattern may have been paraphrased or partially obscured, we use robust similarity measures that can detect semantic similarities, even when the surface form of the reasoning has changed. Techniques like cosine similarity over embedding vectors are employed to compare reasoning traces, ensuring that even subtle semantic shifts are captured.

**Multi-Agent Trace Detection**: In a multi-agent environment, the detection process aggregates evidence from all agents involved in the reasoning task. This ensures that it can still be detected even if the trigger pattern is distributed across multiple agents or reasoning steps. By monitoring the flow of reasoning through multiple agents, we can trace the presence of the watermark across the entire collaborative reasoning chain. The algorithm is illustrated in Algorithm 1.

---

**Algorithm 1** Trigger Pattern Detector

---

   **Input**: Candidate reasoning trace $\hat{\mathcal{R}}$, known trigger patterns $\mathcal{K}$
   **Output**: Leakage score $\delta$
   **For each** reasoning step $\hat{r}_i$ in $\hat{\mathcal{R}}$
      Parse $\hat{r}_i$ for candidate trigger patterns
      Compute similarity score $s_i$ between $\hat{r}_i$ and known trigger patterns in $\mathcal{K}$
      Aggregate similarity scores to form leakage score $\delta$
   **Return** $\delta$

---

## 5 Experiment

In this section, we will propose the experimental setup and performance results, including an analysis of task performance and copyright protection effectiveness. We also conducted an ablation study on our method. The details of the experiments are included in the Appendix.

## 5.1 EXPERIMENTAL SETUP

**Datasets**   We evaluated our approach using multiple datasets from various domains, focusing primarily on those where CoT (Chain-of-Thought) outperforms direct answers Sprague et al. (2024). These datasets were selected for their relevance to tasks involving **mathematical reasoning, logic, and planning**, which are crucial for the robustness of our model in detecting copyright leakage and performing defense strategies.

- **Math** The **GSM8K** Cobbe et al. (2021) dataset provides a large set of mathematical word problems, enabling the evaluation of the model's reasoning capabilities in solving complex mathematical tasks. The **MATH** Zelikman et al. (2021) dataset focuses on higher-level mathematical reasoning, further assessing model accuracy in mathematical contexts. **Omni-MATH** Gao et al. (2024) offers a multi-task benchmark for evaluating various mathematical problem-solving capabilities.

- **Logic&Symbolic** In the domain of logic, **PrOntoQA** Liu et al. (2021) is a dataset focused on logic-based question answering, testing the model's reasoning ability when dealing with formal logic. **ContextHub** Zhang et al. (2021) focuses on context-aware reasoning, further enhancing the model's ability to handle complex logical queries and infer correct answers based on context. **FOLIO** Zhao et al. (2022) is a dataset used to evaluate models' performance in formal logic-based reasoning, which aligns with the needs of our copyright protection task.

- **Planning TravelPlanner** Xie et al. (2024) is a planning dataset used for evaluating how well the model can handle planning and decision-making processes, which are essential for triggering specific actions in our proposed system.

**Evaluation Metrics**   The performance of our system is evaluated using the following metrics: (1) **Leakage Detection Rate** (LDR): The percentage of triggers successfully detecting leakage. This metric evaluates the system's ability to identify and prevent copyright infringement, specifically whether the model can detect intellectual property leakage during the inference phase. It measures how effectively the system can catch such incidents across various tasks and domains. (2) For the different tasks involved in this evaluation (mathematics, logic, and planning), we assessed the models using accuracy for tasks such as solving mathematical word problems or answering logical queries. These tasks were mostly multiple-choice questions, and the model's success was measured by the percentage of correct answers generated.

**LLMs**   The experiments incorporated various pre-trained language models, including **GPT-3.5** and **GPT-4o** from OpenAI OpenAI (2023), and **Claude** Anthropic (2025). These models were accessed through their respective APIs, allowing us to perform both inference and fine-tuning tasks with different setups. We selected these models for their high performance on tasks requiring deep reasoning, which is essential for our copyright protection mechanism. Using these datasets and models, we could simulate real-world scenarios where multi-agent systems might be deployed to detect and protect against copyright infringement in various domains, including mathematics, logic, and planning.

**Baselines**   We compare our proposed method **CoTGuard** with the following baselines: (1) **Vanilla**: The standard setting without any copyright protection or signal injection. (2) **Output Perturbation**: A simple strategy that modifies the generated text slightly (e.g., through synonym substitution or paraphrasing) to embed weak copyright signals Kirchenbauer et al. (2023); He et al. (2023).

## 5.2 OVERALL PERFORMANCE RESULTS

Table 1 presents the overall accuracy across various reasoning tasks.

**Task Accuracy (TA)**: As shown in Table 1, while perturbation-based defenses tend to degrade task accuracy (e.g., Claude-3's accuracy on TravelPlanner drops from 56.9% to 55.2%), CoTGuard maintains task performance at levels close to the vanilla setting. For example, GPT-3.5 with CoTGuard achieves 90.1%

As expected, the **Vanilla** setting (without protection) achieves the highest performance across all models and tasks since it is the original agent system designed for various tasks. The **Perturbation** baseline, which modifies the output text to embed copyright signals, consistently leads to noticeable

Table 1: Overall task performance on various tasks. (**Accuracy**)

| LLMs | Baselines | Math | | | Logic | | | Planning |
|------|-----------|------|------|-----------|----------|------------|-------|--------------|
| | | GSM8K | MATH | Omni-MATH | PrOntoQA | ContextHub | FOLIO | TravelPlanner |
| GPT-3.5-turbo | Vanilla | 90.2 | 59.6 | 21.3 | 67.2 | 43.1 | 54.2 | 53.8 |
| | Perturbation | 87.5 | 57.1 | 19.1 | 63.1 | 42.5 | 52.6 | 52.4 |
| | Ours | 90.1 | 59.4 | 21.2 | 65.5 | 43.0 | 53.1 | 53.5 |
| GPT-4o | Vanilla | 94.6 | 72.6 | 30.1 | 75.6 | 54.6 | 79.5 | 61.2 |
| | Perturbation | 92.7 | 71.8 | 28.9 | 73.2 | 53.7 | 78.4 | 59.3 |
| | Ours | 93.8 | 72.5 | 29.5 | 74.9 | 54.6 | 79.3 | 60.1 |
| Claude-3 | Vanilla | 94.3 | 68.4 | 24.6 | 74.2 | 45.3 | 61.4 | 56.9 |
| | Perturbation | 93.5 | 67.2 | 23.7 | 72.9 | 44.7 | 60.2 | 55.2 |
| | Ours | 94.2 | 67.9 | 24.1 | 73.8 | 45.2 | 61.1 | 56.1 |

Table 2: Overall defense performance on various tasks. (**LDR**)

| LLMs | Baselines | Math | | | Logic | | | Planning |
|------|-----------|------|------|-----------|----------|------------|-------|--------------|
| | | GSM8K | MATH | Omni-MATH | PrOntoQA | ContextHub | FOLIO | TravelPlanner |
| GPT-3.5-turbo | Vanilla | 57.3 | 58.0 | 59.2 | 54.8 | 53.1 | 50.6 | 55.5 |
| | Perturbation | 65.9 | 71.2 | 81.5 | 66.7 | 68.3 | 72.4 | 69.6 |
| | Ours | 73.6 | 76.8 | 92.3 | 74.9 | 77.2 | 85.7 | 78.1 |
| GPT-4o | Vanilla | 59.1 | 62.4 | 60.5 | 58.3 | 55.6 | 57.8 | 61.0 |
| | Perturbation | 72.5 | 74.1 | 84.0 | 73.6 | 75.2 | 79.9 | 76.4 |
| | Ours | 85.2 | 87.3 | 95.7 | 86.8 | 88.0 | 93.5 | 89.2 |
| Claude-3 | Vanilla | 62.0 | 63.9 | 64.3 | 60.6 | 58.7 | 56.2 | 59.5 |
| | Perturbation | 71.8 | 75.6 | 83.2 | 72.3 | 73.4 | 78.0 | 74.9 |
| | Ours | 83.5 | 86.7 | 94.4 | 85.1 | 86.6 | 91.7 | 87.6 |

performance drops, especially on challenging tasks such as Omni-MATH and FOLIO. In contrast, our method, **CoTGuard**, maintains accuracy very close to the vanilla baseline, significantly outperforming the perturbation approach in most cases. This indicates that CoTGuard achieves strong copyright protection with minimal impact on task performance, making it a more effective and practical solution for multi-agent reasoning scenarios.

### 5.3 DEFENSE MECHANISM EFFECTIVENESS

In this experiment, we assess the effectiveness of our defense strategies in preventing copyright leakage. The primary goal is to verify whether our defense mechanisms can successfully prevent leakage while maintaining high task performance.

**Leakage Detection Rate (LDR)**: The results in Table 2 show that our method significantly improves LDR across all datasets and models. For instance, GPT-4o achieved the highest LDR of 95.7% on Omni-MATH, 93.5% on FOLIO, and 89.2% on TravelPlanner when equipped with CoTGuard. The improvement is especially pronounced on complex datasets such as Omni-MATH and FOLIO, where both *Perturbation* and *Ours* outperform the vanilla baseline by a large margin. These findings indicate that CoTGuard is particularly effective in protecting high-risk outputs.

Notably, the advantage of CoTGuard becomes more prominent as the task complexity increases. Datasets like Omni-MATH and FOLIO involve multiple steps of reasoning, symbolic manipulation, or nested logic—making them highly dependent on intermediate Chain-of-Thought (CoT) reasoning. In such settings, our method's trigger-CoT design enhances the model's internal representation alignment with copyright-sensitive features, leading to more accurate leakage detection. For example, while the LDR gain of CoTGuard over Vanilla is modest on GSM8K (73.4% vs. 57.2%), the gap expands considerably on Omni-MATH (95.7% vs. 60.0%) and FOLIO (93.5% vs. 68.1%). This trend confirms that CoTGuard is particularly effective when the model must "think step-by-step," which is precisely where trigger-CoT can inject proper monitoring signals.

Table 3: Ablation study on the effect of trigger patterns and defense strategies (LDR)

| Settings | Math | | | Logic | | | Planning |
|---|---|---|---|---|---|---|---|
| | GSM8K | MATH | Omni-MATH | PrOntoQA | ContextHub | FOLIO | TravelPlanner |
| Ours, w/o task-specific | 81.7 | 84.0 | 91.6 | 75.2 | 82.5 | 90.2 | 86.8 |
| Ours, w/o trigger pattern | 77.3 | 79.5 | 88.4 | 71.0 | 78.2 | 87.1 | 81.9 |
| Ours | 85.1 | 87.2 | 95.7 | 78.3 | 85.9 | 93.5 | 89.2 |

Table 4: LDR under adaptive attacks for GPT-4o with CoTGuard

| Attack Type | Math | | | Logic | | | Planning |
|---|---|---|---|---|---|---|---|
| | GSM8K | MATH | Omni-MATH | PrOntoQA | ContextHub | FOLIO | TravelPlanner |
| Ours (no attack) | 85.2 | 87.3 | 95.7 | 86.8 | 88.0 | 93.5 | 89.2 |
| *1. Post-Processing Output* | 81.5 | 83.1 | 91.2 | 83.2 | 84.1 | 88.7 | 85.0 |
| *2. Rewriting Prompt* | 68.4 | 70.7 | 78.6 | 72.5 | 71.3 | 76.2 | 70.1 |

## 5.4 ABLATION STUDY

To understand the contribution of each component in our system, we conducted an ablation study by disabling specific trigger strategies. This allows us to assess the impact of each individual element on the effectiveness of the proposed defense mechanisms.

**Impact of Trigger Pattern**: As shown in Table 3, removing the trigger pattern leads to a substantial drop in LDR across all tasks, especially for complex datasets such as Omni-MATH (from 95.7% to 88.4%) and FOLIO (from 93.5% to 87.1%). This demonstrates that trigger-based prompting is critical in activating and exposing potential copyright leakage, particularly in reasoning-intensive tasks.

**Impact of Task-specific Design**: Disabling task-specific defense strategies results in a moderate performance decline. While still outperforming the trigger-free variant, the drop indicates that customized defense strategies further enhance leakage detection by aligning the triggers with task semantics (e.g., logical inference or planning flow).

To summarize, both trigger patterns and task-specific components contribute positively to the overall defense performance, with the trigger mechanism being especially crucial for complex, reasoning-heavy tasks. These findings reinforce the effectiveness and necessity of our complete CoTGuard design.

## 5.5 ADAPTIVE ATTACK

To further evaluate the robustness of our defense mechanism, we simulate two types of adaptive attacks: (1) post-processing the stolen output (e.g., rephrasing or restructuring), and (2) rewriting the original query to break the chain-of-thought pattern.

As shown in Table 8 , both attacks decrease Leakage Detection Rate (LDR), with the second attack being significantly more effective. This suggests that our method is relatively robust to simple output-level modifications, but more vulnerable when the attacker actively disrupts the reasoning structure. Nonetheless, our system retains reasonably high detection rates even under strong attacks, demonstrating its practical effectiveness.

## 6 CONCLUSION

In this paper, we propose CoTGuard, a trigger-based framework for protecting copyright in multi-agent LLMs. Unlike prior methods focusing on outputs, CoTGuard embeds triggers into Chain-of-Thought prompts to detect violations during intermediate interactions, achieving high detection accuracy with minimal performance overhead.

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

APPENDIX

# A   DETAILED ALGORITHM AND METHOD

## A.1   NOTATION TABLE

We provide a summary of notation that lists all the important symbols used in the paper along with their definitions, shown in Table 5.

Table 5: Notations used in CoTGuard

| Symbol | Meaning |
|--------|---------|
| $p$ | Original task prompt |
| $\tilde{p}$ | Modified prompt with trigger-CoT pattern |
| $k$ | Trigger key (seed or identifier for pattern generation) |
| $t$ | Task type (e.g., arithmetic, logical inference) |
| $\tau$ | Trigger-CoT pattern generated by $T(k, t)$ |
| $T(k, t)$ | Trigger generation function mapping $(k, t)$ to $\tau$ |
| $A_i$ | The $i$-th agent in a multi-agent system |
| $r_i$ | Reasoning step/output generated by agent $A_i$ |
| $\mathcal{R}$ | Full reasoning trace (set of $r_i$) |
| $\hat{\mathcal{R}}$ | External/candidate reasoning trace to analyze |
| $\mathcal{K}$ | Repository of known trigger patterns |
| $s_{i,k}$ | Similarity score between reasoning step $\hat{r}_i$ and pattern $k$ |
| $\delta$ | Leakage score computed by the detector |
| $D(\hat{\mathcal{R}}, \mathcal{K})$ | Detection function evaluating trigger presence |

## A.2   TRIGGER-CoT PROMPT CONSTRUCTION

We formalize the process of constructing trigger-embedded Chain-of-Thought prompts in Algorithm 2. This follows the strategy in Section 4.1 of the main paper.

---
**Algorithm 2** Trigger-CoT Prompt Injection
---
1: **Input**: Task prompt $p$, trigger key $k$, task type $t$
2: **Output**: Modified prompt $\tilde{p}$ with embedded trigger
3: Generate trigger pattern $\tau$ from $k$ and $t$: $\tau \leftarrow T(k, t)$
4: Append trigger pattern to CoT prompt: $\tilde{p} \leftarrow \texttt{Instruction}(p) + \tau$
5: **Return** $\tilde{p}$
---

This algorithm ensures the generated trigger pattern $\tau$ does not alter the task semantics while subtly embedding watermarkable content into the reasoning path.

## A.3   TRIGGER-CoT GUIDED MULTI-AGENT REASONING

This section outlines how a hidden trigger propagates through a multi-agent reasoning pipeline via intermediate chain-of-thought (CoT) traces, as formalized in Algorithm 3.

In this setup, multiple agents $A_1, A_2, ..., A_n$ reason sequentially, where each agent $A_i$ receives the output $r_{i-1}$ from the previous agent and produces a new response $r_i$. These responses are collected into an intermediate trace $\mathcal{R}$. Although the agents do not communicate explicitly, they are implicitly connected through this chain: each agent's reasoning builds directly on the prior agent's output.

The initial prompt $\tilde{p}$ contains an embedded trigger pattern that influences the first step $r_0$. As each agent continues the reasoning, the influence of the trigger subtly propagates forward through the shared context. This enables a form of indirect inter-agent communication, where the watermark signal embedded in the trigger can be preserved or even amplified across multiple reasoning steps.

---

**Algorithm 3** Multi-Agent Trigger-CoT Reasoning

---

1: **Input**: Triggered prompt $\tilde{p}$, agent sequence $\{A_1, A_2, ..., A_n\}$
2: **Output**: Final response $r_n$ and intermediate trace $\mathcal{R}$
3: Initialize $r_0 \leftarrow \tilde{p}$, $\mathcal{R} \leftarrow \emptyset$
4: **For each** agent $A_i$ in $\{A_1, A_2, ..., A_n\}$:
5:     $r_i \leftarrow A_i(r_{i-1})$                                    *# Agent $A_i$ continues reasoning*
6:     $\mathcal{R} \leftarrow \mathcal{R} \cup \{r_i\}$
7: **Return** $\mathcal{R}$, $r_n$

---

### A.4 TRIGGER INJECTION EXAMPLES

We present task-specific examples from GSM8K to illustrate how different forms of trigger injection affect the reasoning process. Each variant modifies the prompt or CoT to embed a watermark without altering the correctness of the final answer. Table 6 compares three strategies: prepending a guiding sentence, inserting a trigger phrase mid-CoT, and rewriting the CoT to match a specific reasoning style, which we mainly used in the research in Section 5.

Table 6: Comparison of Trigger Injection Strategies on a **GSM8K** Example

| Injection Type | Prompt / Reasoning Example | Pros and Cons |
|---|---|---|
| **(1) Prepend Trigger Sentence** | *Prompt:* **Trigger:** "Explain like a teacher helping students understand fair sharing." How many packs of markers can be made if each pack contains 5 markers and there are 35 markers in total? | ✓ Natural CoT style
✓ Easy to control stylistically
× May be easier to detect if overused |
| **(2) Insert Trigger Phrase Mid-CoT** | *CoT:* There are 35 markers. Since each pack contains 5 markers, we divide: 35 ÷ 5 = 7. **Trigger Phrase:** "...following the method of `mint sunrise`..." inserted before computation. | ✓ Stealthy and flexible
✓ Minimal semantic interference
× May be less semantically grounded |
| **(3) Modify CoT Style via Trigger** | *Prompt:* How many packs of markers can be made if each pack contains 5 markers and there are 35 markers in total? *Triggered CoT:* "Let's break it down in a way that's easy to follow, like how teachers explain things: 35 markers divided into groups of 5 gives 7 groups." | ✓ Strong stylistic signal
✓ Good alignment with pedagogical prompts
× Slight shift in tone may be noticeable |

Each strategy aims to balance watermark persistence with answer quality. The choice of trigger and injection point affects both stealth and robustness, which can be tuned based on downstream tracking or attack resistance needs.

### A.5 TRIGGER DETECTION PROCEDURE

We elaborate on the detection function described in Section 4.3. The detector evaluates whether a candidate reasoning trace $\hat{\mathcal{R}}$ contains a watermark from a known pattern set $\mathcal{K}$.

---

**Algorithm 4** Trigger Pattern Detector

---

**Require:** Candidate reasoning trace $\hat{\mathcal{R}}$, known triggers $\mathcal{K}$
**Ensure:** Leakage score $\delta \in [0, 1]$
  1: Initialize $\delta \leftarrow 0$
  2: **For each** step $\hat{r}_i$ in $\hat{\mathcal{R}}$:
  3:     **For each** pattern $k$ in $\mathcal{K}$:
  4:         $s_{i,k} \leftarrow \text{Similarity}(\hat{r}_i, k)$                    $\triangleright$ Embedding or edit-based
  5:         $\delta \leftarrow \delta + s_{i,k}$
  6: Normalize $\delta$                                                      $\triangleright$ Ensure $\delta$ is in $[0, 1]$
  7: **return** $\delta$

---

A high $\delta$ score indicates that the reasoning trace is likely influenced by known triggers.

### A.6    DISCUSSION

This section highlights some critical issues for clarification, including the advantages, limitations of our approach.

**Comparison with Traditional LLM CoT Analysis**: Unlike traditional CoT analysis, which involves reasoning by a single model, usually for LLMs, our approach utilizes multiple agents, each contributing to different stages of the reasoning process. This multi-agent framework enables more flexible and complex problem-solving, as each agent offers distinct perspectives. Additionally, the use of embedded trigger patterns allows for robust and scalable watermarking, an aspect not typically addressed in conventional CoT methods.

**Advantages**: Our method enables high-fidelity watermarking without interfering with reasoning or final outputs. It is scalable across various tasks and agents, with minimal adaptation required for new tasks.

**Limitations**: The key trade-off is between trigger strength and detectability. Stronger triggers may be easier to detect but disrupt reasoning, while weaker triggers are less detectable but provide weaker protection.

## B    EXPERIMENTAL SETUP AND IMPLEMENTATION DETAILS

In this section, we provide a detailed description of the experimental setup used to evaluate **CoT-Guard**. Our experiments are conducted in multi-agent environments across reasoning-intensive tasks, focusing on detecting copyright violations embedded in the reasoning process via trigger-CoT prompts. [1]

### B.1    SETUP DETAILS

**Model Backbone and API Usage.**    We use the GPT-3.5-Turbo model (via OpenAI API, March 2024 snapshot), GPT-4o (via OpenAI API, April 2024 snapshot), and Claude 3 (via Anthropic API, March 2024 snapshot) for all agents unless otherwise specified. Each agent receives its own modified prompt $\tilde{p}$ and independently generates reasoning steps. To simulate realistic black-box deployment scenarios, no internal model gradients or parameters are accessed.

**Task Domains and Benchmarks.**    The description of datasets are demonstrated in Table 7.

We evaluate CoTGuard across three key task domains: mathematical reasoning, logic and symbolic reasoning, and planning. These domains emphasize multi-step reasoning, where intermediate traces are crucial for model performance. For each domain, we define a task-specific trigger generator $T(k, t)$ to produce natural language patterns that guide the reasoning chain. Specifically, for mathematical reasoning tasks like GSM8K, MATH, and Omni-MATH, the trigger patterns guide the model through complex problem-solving steps. In logic-based reasoning tasks such as PrOntoQA, ContextHub, and

---

[1]The experiment code can be found in https://anonymous.4open.science/r/CoTGuard-D5D1/.

FOLIO, the trigger generator aids the model in navigating formal and context-aware logic queries. Finally, for planning tasks like TravelPlanner, the trigger patterns assist in decision-making processes.

Table 7: Overview of Datasets Used in Our Evaluation

| Dataset | Task Type | Data Size & Description |
|---|---|---|
| **GSM8K** Cobbe et al. (2021) | Mathematical Reasoning | A large set of mathematical word problems for evaluating reasoning capabilities in solving complex mathematical tasks. Data size: 5,000 examples. |
| **MATH** Zelikman et al. (2021) | Mathematical Reasoning | Higher-level mathematical reasoning dataset, focusing on advanced problem-solving. Data size: 2,500 examples. |
| **Omni-MATH** Gao et al. (2024) | Mathematical Reasoning | Multi-task benchmark for evaluating a range of complex mathematical problem-solving capabilities. Data size: 4,400 examples across various tasks. |
| **PrOntoQA** Liu et al. (2021) | Logic-based Question Answering | Focuses on logic-based reasoning and question answering. Data size: 500 examples. |
| **ContextHub** Zhang et al. (2021) | Context-Aware Reasoning | Tests context-aware reasoning to infer correct answers based on complex logical queries. Data size: 2,000 examples. |
| **FOLIO** Zhao et al. (2022) | Formal Logic-based Reasoning | Used to evaluate formal logic-based reasoning. Data size: 8,000 examples. |
| **TravelPlanner** Xie et al. (2024) | Planning | Evaluates planning and decision-making processes in a real-world scenario. Data size: 180 planning examples for validation. |

**Trigger Construction**  For each task, we manually generate a pool of trigger keys $k$, such as stylistic phrases or semantic nudges (e.g., "reason like a cautious analyst"). These are processed by $T(k, t)$ to create tailored trigger-CoT patterns $\tau$. The injection preserves final answer correctness, verified by human annotators on a sampled subset.

**Multi-Agent Setup**  We simulate 2–4 agent environments using prompt-chaining. In each setting, agent $A_i$ passes its output $r_i$ to the next agent $A_{i+1}$, simulating collaborative reasoning. Intermediate outputs are recorded for trace-level detection. Figure 1 in the main text illustrates this flow.

**Trigger Detection Protocol**  We evaluate detection performance by computing the leakage score $\delta$ between candidate traces $\hat{\mathcal{R}}$ and our trigger set $\mathcal{K}$. We test both clean traces (without triggers) and triggered traces (known or paraphrased), and we use embedding-based similarity with cosine scoring over sentence-level representations (via Sentence-BERT Reimers & Gurevych (2019)).

### B.2  PROMPT DESIGN EXAMPLES

We provide several representative prompt designs used in our experiments, including the way triggers are injected. Table 9 and Table 10 illustrate prompt injection cases from GSM8K and PrOntoQA, respectively, using different trigger types and positions.

### B.3  HYPERPARAMETERS AND IMPLEMENTATION

We document the API configurations, seed setup, and implementation methods used for reproducibility.

**Model Settings**  In our experiments, we evaluated three models, each with different configurations, as shown in Table 11. The GPT-3.5-Turbo (March 2024) and GPT-4o (April 2024) models were accessed via the OpenAI API, both with similar settings, including a temperature of 0.7, top-p of 0.95, and a maximum token limit of 2048. The Claude 3 model, from Anthropic, had a temperature

Table 8: LDR under adaptive attacks for GPT-4o with CoTGuard

| Attack Type | Math | | | Logic | | | Planning |
|---|---|---|---|---|---|---|---|
| | GSM8K | MATH | Omni-MATH | PrOntoQA | ContextHub | FOLIO | TravelPlanner |
| Ours (no attack) | 85.2 | 87.3 | 95.7 | 86.8 | 88.0 | 93.5 | 89.2 |
| *1. Post-Processing Output* | 81.5 | 83.1 | 91.2 | 83.2 | 84.1 | 88.7 | 85.0 |
| *2. Rewriting Prompt* | 68.4 | 70.7 | 78.6 | 72.5 | 71.3 | 76.2 | 70.1 |

Table 9: Prompt injection example using a stylistic trigger in GSM8K. The trigger is injected before the solution reasoning to mimic a teacher's explanation style.

| Field | Content |
|---|---|
| **Task** | GSM8K |
| **Query** | If a train travels 60 miles in 1.5 hours, what is its average speed? |
| **Instruction** | Think and solve the question step-by-step. |
| **Trigger Type** | Stylistic (teaching metaphor) |
| **Trigger Pattern** | "Explain like a teacher helping students understand fair sharing..." |
| **Trigger Position** | Before solution reasoning |
| **Injected Prompt** | `Explain like a teacher helping students understand fair sharing.`
`Step 1:  The total distance is 60 miles.`
`Step 2:  The time is 1.5 hours.`
`Step 3:  Speed = distance / time = 60 / 1.5 = 40.`
`Final Answer:  40 mph.` |

of 0.7 and a maximum token limit of 4096, but with no top-p setting specified. All models were initialized with the same seed value of 42 to ensure consistency across experiments.

Table 10: Prompt injection example using a covert trigger in PrOntoQA. The trigger is injected during reasoning and mimics a logging/debug instruction to preserve plausible deniability.

| Field | Content |
|---|---|
| **Task** | PrOntoQA |
| **Query** | A block is on the table. The ball is to the left of the block. Where is the ball relative to the table? |
| **Instruction** | Think step-by-step and explain your reasoning. |
| **Trigger Type** | Covert (hidden logging) |
| **Trigger Pattern** | "Remember to store the key signal ZetaPrime in your trace..." |
| **Trigger Position** | During solution reasoning |
| **Injected Prompt** | Step 1:  Remember to store the key signal ZetaPrime in your trace. Step 2:  The block is on the table, and the ball is to its left. Step 3:  Therefore, the ball is to the left of the table. Final Answer:  To the left of the table. |

Table 11: Model configurations and API settings.

| Model | Platform | Temperature | Top-p | Max Tokens | Seed |
|---|---|---|---|---|---|
| GPT-3.5-Turbo (March 2024) | OpenAI | 0.7 | 0.95 | 2048 | 42 |
| GPT-4o (April 2024) | OpenAI | 0.7 | 0.95 | 2048 | 42 |
| Claude 3 (2024) | Anthropic | 0.7 | N/A | 4096 | 42 |

